# Macroscopic control of cell electrophysiology through ion channel expression

**Mario García-Navarrete†, Merisa Avdovic†, Sara Pérez-Garcia†, Diego Ruiz Sanchis, Krzysztof Wabnik\***

Centro de Biotecnología y Genómica de Plantas (Universidad Politécnica de Madrid – Instituto Nacional de Investigación y Tecnología Agraria y Alimentaria), Pozuelo de Alarcón, Spain

**\*For correspondence:**
k.wabnik@upm.es

†These authors contributed equally to this work

**Competing interest:** The authors declare that no competing interests exist.

**Abstract** Cells convert electrical signals into chemical outputs to facilitate the active transport of information across larger distances. This electrical-to-chemical conversion requires a tightly regulated expression of ion channels. Alterations of ion channel expression provide landmarks of numerous pathological diseases, such as cardiac arrhythmia, epilepsy, or cancer. Although the activity of ion channels can be locally regulated by external light or chemical stimulus, it remains challenging to coordinate the expression of ion channels on extended spatial–temporal scales. Here, we engineered yeast *Saccharomyces cerevisiae* to read and convert chemical concentrations into a dynamic potassium channel expression. A synthetic dual-feedback circuit controls the expression of engineered potassium channels through phytohormones auxin and salicylate to produce a macroscopically coordinated pulses of the plasma membrane potential. Our study provides a compact experimental model to control electrical activity through gene expression in eukaryotic cell populations setting grounds for various cellular engineering, synthetic biology, and potential therapeutic applications.

## Editor's evaluation

The important contribution of this study is the ability to leverage engineered gene circuits to control cellular membrane potential. The presentation of the data in this work is convincing and the controls are in place to demonstrate that electrophysiological changes arise from external chemical stimuli. This study will be of interest to those working on non-neuronal bioelectricity, particularly synthetic biologists and bioengineers.

## Introduction

Electrical signals provide an active mechanism for the rapid delivery of information across noisy cells and tissues. Prominent examples of this phenomenon across kingdoms include excitable neuronal circuits (*Aron and Yankner, 2016*), plant defense signaling (*Masatsugu, 2018*), and metabolic coordination of biofilm growth (*Prindle et al., 2015*). These seemingly different forms of electrical signaling involve ion channels. Outward-rectifying potassium channels release potassium from the intracellular reservoir to the extracellular space, thereby allowing for potassium exchange between neighboring cells (*Debanne et al., 1997*). A rapid gating of ion channels at the subcellular level maintains the balance in the plasma membrane potential (PMP), which is central to cell electrophysiology (*Naundorf et al., 2006*).

The activity of potassium channels can be locally modulated by voltage-gating, mechanical or light stimulus, and external ligands. In the last decade, advances in optogenetics and chemical biology of ion channels have powered numerous medical applications through local regulation of electrical activity in living cells with the ultimate goal of treating major life-threatening diseases (*Gradinaru et al., 2010*; *Häfner and Sandoz, 2022*; *Snyder, 2017*; *Montnach et al., 2022*). Interestingly, recent studies indicate the spatial–temporal regulation of ion channel expression and membrane potential status is critical for landmarking pathological conditions such as cardiac arrhythmia, epilepsy, or various types of cancer (*Rosati and McKinnon, 2004*; *Lastraioli et al., 2015*; *Zsiros et al., 2009*; *Niemeyer et al., 2001*; *Biasiotta et al., 2016*). However, a major challenge is to achieve rational control of ion channels on extended spatial–temporal scales. Such a strategy would provide a basis foundation for advanced applications in treating epilepsy, chronic pain, irregular heartbeats, or potentially various types of cancer. Nevertheless, there is a lack of experimental models allowing a guided modulation of eukaryotic cell electrophysiology through modulation of ion channel expression.

To address this challenge, we build a synthetic gene regulatory mechanism that is capable of controlling ion channel expression in the cell populations of the model eukaryote *Saccharomyces cerevisiae* based on environmental changes. By combining live-cell imaging in microfluidic devices with computer modeling we tested a suitable eukaryotic model for macroscopic real-time modulation of ion channels and PMP in living cell collectives.

## Results and discussion

### Controlling macroscopic ion channel expression through plant hormones

Chemical messengers or light can selectively control local activity of ion channels on the plasma membrane (*Gradinaru et al., 2010*; *Häfner and Sandoz, 2022*; *Snyder, 2017*; *Montnach et al., 2022*). In contrast, we sought to implement an alternative system for ion channel modulation by chemically coordinating the ion channel expression at the macroscopic level. To test this concept, we used a model eukaryote yeast *S. cerevisae*. Previously, we have developed a synthetic two component circuit to control gene expression across cell populations based on chemical stimulation with phytohormones auxin and salicylate (*Pérez-García et al., 2021*). This circuit is composed of engineered Mar-type bacterial regulator (*Will and Fang, 2020*); IacR transcriptional activator and MarR transcriptional repressor (*Pérez-García et al., 2021*; *Figure 1A*), that are inhibited by auxin (IAA) and salicylate (SA), respectively. We sought this particular circuit is ideal for our application since it can coordinate gene expression in yeast on extended spatiotemporal scale (*Pérez-García et al., 2021*).

It is known that the overexpression of potassium channel TOK1 in yeast leads to PMP hyperpolarization (*Sesti et al., 2001*). Therefore, in theory, our strategy would allow a direct control of ion channel activity through coordinated modulation of gene expression levels, leading to global alterations of PMP. In our system changes in channel expression would cause associated changes in PMP primarily based on the temporal status of phytohormones in the environment (*Figure 1A*). Importantly, a voltage-gated channel opening, dependent on internal metabolic processes, could be attenuated by the use of constitutively open ion channel (*Figure 1B*). Thus, we used an open bacteria potassium channel KcsA* (*Doyle et al., 1998*; *Cuello et al., 2010*; *Sun et al., 2020*) as the output module of the synthetic circuit (*Figure 1B*, *Figure 1—figure supplement 1*). To create a dynamic environment, we employed controlled conditions on the microfluidic chip where yeast can grow under continuous media perfusion stimulated with antithetic pulses of SA and IAA with the regulable frequency (*Figure 1C*). To monitor PMP changes we have tested several commonly used dyes in yeast such as DIBAC4(3), DIS-C3(3), and cationic dye Thioflavin T (ThT) (*Peña et al., 2020*) and we found that a ThT outperforms other dyes in terms of stability, fluorescence level, low hydrophobicity (*Peña et al., 2020*), and near-linear response to potassium clamp experiments (*Figure 1—figure supplement 2A–C*). Finally, all these features of ThT were ideal for the long-term microfluidics experiments when stability, fluorescence, and low absorption in PDMS are critical.

Under coordinated changes of IAA and SA in the environment, the control strain that lacks the synthetic circuit (*Figure 1A*) does not show any regular changes in ThT fluorescence over time (*Figure 1D, E* and *Figure 1—figure supplement 3A*, *Figure 1—video 1*). Whereas the open-loop circuit driving KcsA channel in yeast (*Figure 1A, B*) showed noisy but recognizable fluctuations of ThT

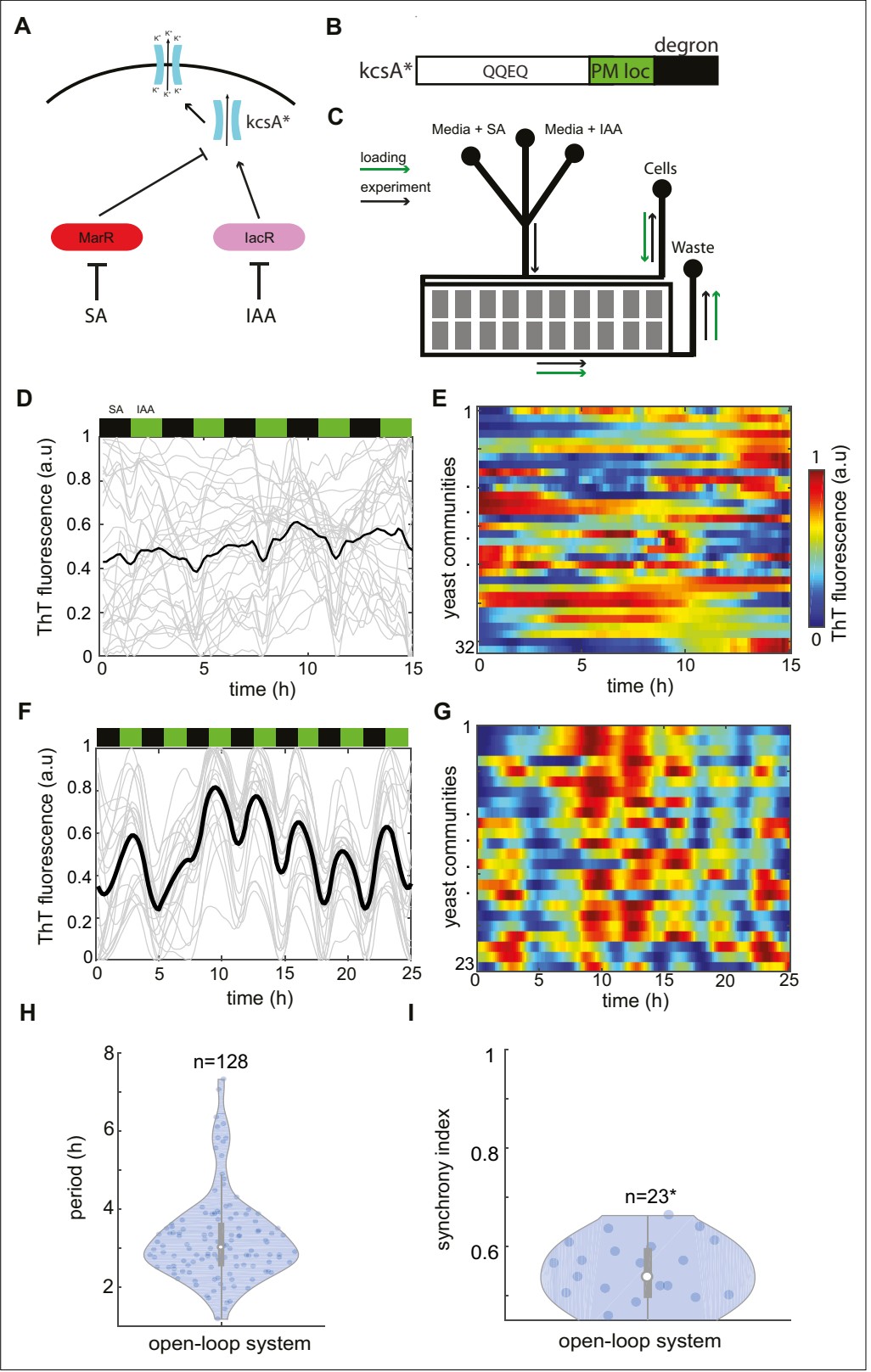

**Figure 1.** Control of ion channel expression through open-loop circuit. (**A**) Schematic of open-loop system driving expression of constitutively open KcsA* bacterial potassium channel. Activator IacR and repressor MarR (tagged with ODC degron; *Takeuchi et al., 2008*) are induced by galactose and repressed by auxin and salicylate, respectively (*Pérez-García et al., 2021*). (**B**) KcsA* contains plasma membrane localization (PM loc, green) and

*Figure 1 continued on next page*

*Figure 1 continued*

c-terminus mouse ODC degron (degron, black) sequences. (**C**) Schematic of microfluidic device used in the study. Flow channels widths were 120 µm for in the mixer module and 500 µm in main channels. The approximate height of channels was 25 µm. Cell traps had 500 × 500 µm size and height of approximately 7 µm. Loading direction is marked with green arrows, while anticipated flow in the experiment with black arrows. Time traces (**D**) and heat map (**E**) of Thioflavin T (ThT) fluorescence in control yeast strain do not show any organized features under 3-hr periodic phytohormone stimuli. (**F–I**) ThT fluorescence changes in the open-loop circuit (**A**) under 3-hr periodic phytohormone stimuli show distinct fluctuations (**F, G**), yet broad period distribution (**H**) and low synchronicity of ~50% (**I**). Periods of peaks were measured for all measured trapping regions without averaging as shown in (**H**). (**I**) Synchrony index is measured for all *n* = 23 trapping regions (communities) each containing ~10,000 yeast cells. Positions of phytohormone stimuli are shown with black (SA) and green (IAA) boxes above the time traces. Each technical experiment has been repeated at least two times with similar results. Violin plots represent medians (white dots), interquartile zones (gray bars) and 95% confidence levels (solid gray line).

The online version of this article includes the following video, source data, and figure supplement(s) for figure 1:

**Source data 1.** Source data for *Figure 1H, I*.

**Figure supplement 1.** The plasmids used for open-loop circuit.

**Figure supplement 2.** Thioflavin T (ThT) cationic dyes show linear response to external potassium applications and high stability compared to other plasma membrane potential (PMP) dyes.

**Figure supplement 2—source data 1.** Source data for *Figure 1—figure supplement 2A–C*.

**Figure supplement 3.** Autocorrelation and peak analysis of Thioflavin T (ThT) traces in control and open-loop system.

**Figure supplement 3—source data 1.** Source data for *Figure 1—figure supplement 3C–E*.

**Figure 1—video 1.** The absence of Thioflavin T (ThT) oscillations in the control strain.
https://elifesciences.org/articles/78075/figures#fig1video1

**Figure 1—video 2.** Weakly coupled Thioflavin T (ThT) oscillations in the strain integrating the open-loop synthetic circuit.
https://elifesciences.org/articles/78075/figures#fig1video2

---

fluorescence with weak coupling between colonies detected by cumulative autocorrelation analysis (*Figure 1F, G*, *Figure 1—figure supplement 3B*, *Figure 1—video 2*). The period of ThT fluorescence showed a broad distribution around the phytohormone stimuli period (*Figure 1H* and *Figure 1—figure supplement 3C*). Also, there was a substantial variability in amplitudes (*Figure 1—figure supplement 3E*) but less of peak widths (*Figure 1—figure supplement 3D*). To complement autocorrelation analysis, we developed a quantitative metric of 'synchrony index' defined as $1R$ where $R$ is the ratio of differences in subsequent ThT peak positions among cell communities (phase) to expected period. This metrics describes how well are yeast colonies synchronized with each other under guidance of the common environmental cue. Notably, we found that around 50% yeast colonies show synchronicity of ThT fluorescence (*Figure 1I*). Based on these data, we concluded that this open-loop system does not provide sufficiently robust changes in PMP to guide electrical activity on the macroscopic scale.

## Encoding dual-feedback regulation increases speed and robustness of response in yeast collectives

To further improve the performance of our system we sought to design a closed-loop feedback circuit that could encode features such as fast responsiveness (due to feedback) and demonstrate noise filtering capability of excitable systems (*Lindner et al., 2004*). For that we coupled lacR and MarR through positive and negative feedback loops in the dual-feedback synthetic gene circuit (*Figure 2A*). To analyze the robustness and dynamics of the designed circuit we first performed computer model simulations of this lacR–MarR feedback system and identified regimes that demonstrate the prominence for the excitable dynamics (*Figure 2B* and *Figure 2—figure supplement 1*). Model predictions revealed minimal prerequisites for the transit between steady state (*Figure 2—figure supplement 1A*), excitability (*Figure 2B*, *Figure 2—figure supplement 1B*), and oscillations (*Figure 2—figure supplement 1C*). These conditions directly relate to the ratio of lacR–MarR deactivation which depends primarily on IAA and SA changes (*Figure 2D*, *Figure 2—figure supplement 1A–C*).

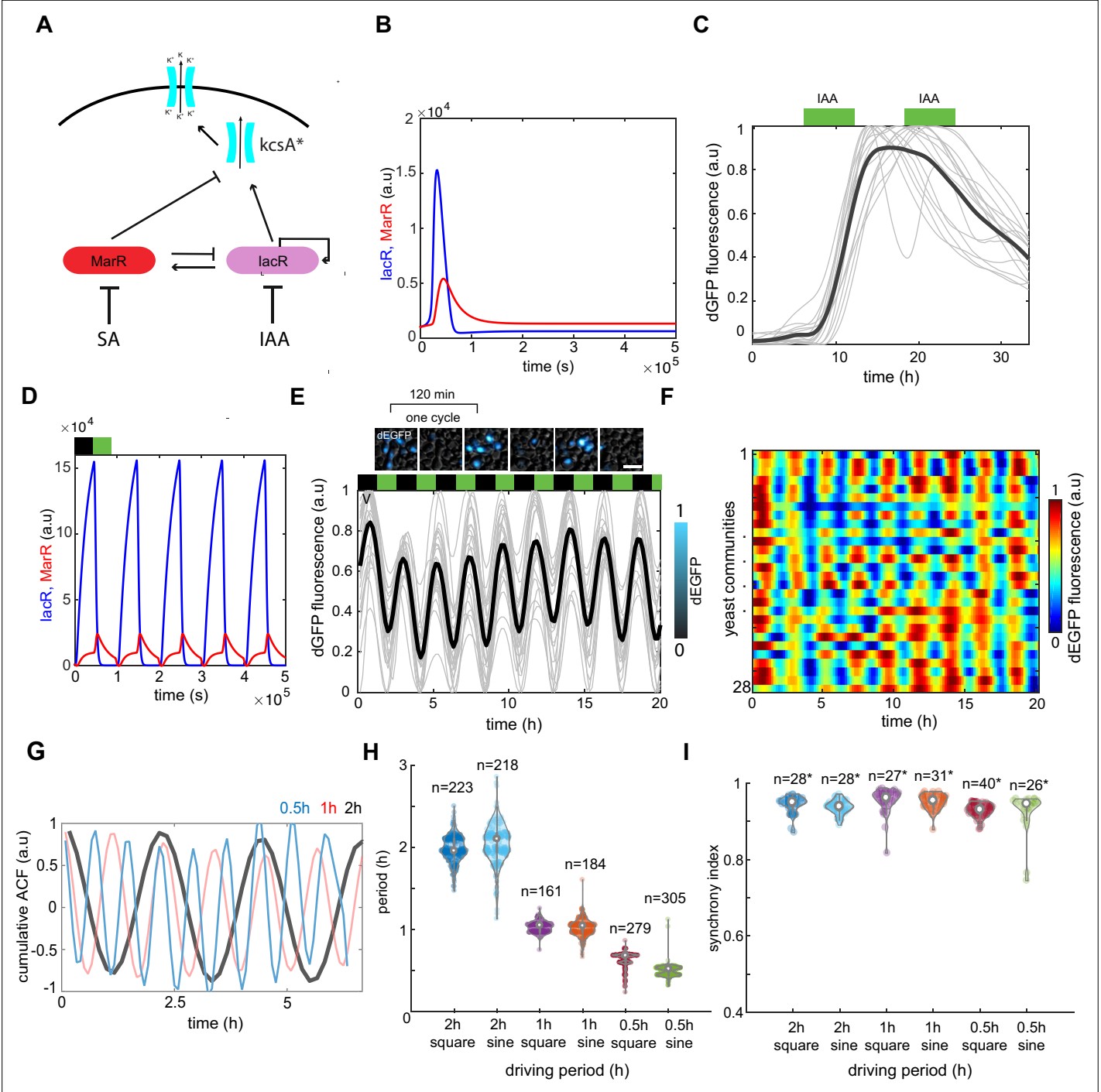

**Figure 2.** A synthetic dual-feedback circuit coordinates ion channel expression in yeast communities. (**A**) Schematic of dual-feedback circuit controlling downstream expression of bacterial potassium channel KcsA*. All components of the system are controlled by the same promoter to allow integration of both positive and negative feedbacks. (**B**) Computer model simulations of a synthetic dual-feedback circuit in the excitable regime. (**C**) Checking for a refractory period in dEGFP circuit reporter response under two long 12-hr pulses of auxin (green boxes). Time-lapse traces from microfluidic device are shown with the mean trend (black curve). Note inability of cell to respond to second pulse of IAA. (**D**) Computer model simulation of a circuit subjected to series of antithetic pulses of SA (black box inability) and IAA (green box). (**E**) Experimental time traces of dEGFP fluorescent marker under 2-hr antithetic pulses of SA and IAA. Note consistency of period between different yeast colonies and some level of variability in peak amplitudes. (**F**) Heat maps of time traces for $n = 28$ yeast communities reported simultaneously in the microfluidic device. (**G**) Cumulative autocorrelation trends for three different stimuli periods (2 hr [black], 1 hr [red], and 30 min [blue]). (**H**) Violin plot shows the consistency of dEGFP period for different stimuli shape and period. (**I**) Violin plot of synchrony index that was measured among all yeast communities (each containing ~10,000 cells) for different stimuli shape and

*Figure 2 continued on next page*

*Figure 2 continued*

period. Each experiment has been repeated at least two times with similar results. Violin plots represent medians (white dots), interquartile zones (gray bars) and 95% confidence levels (solid gray line).

The online version of this article includes the following video, source data, and figure supplement(s) for figure 2:

**Source data 1.** Source data for *Figure 2H, I*.

**Figure supplement 1.** Computer model simulations of IacR–MarR feedback circuit.

**Figure supplement 2.** Plasmids used to construct dual-feedback circuit and engineered ion channels.

**Figure supplement 3.** Excitable control of circuit output with the pair of phytohormones.

**Figure supplement 4.** Time-lapse synthetic circuit characterization using high-throughput fluorescence assays.

**Figure supplement 5.** The frequency of phytohormone stimuli determines the dynamics of the reporter gene under different environmental drivers.

**Figure supplement 6.** Analysis of response characteristics under different environments.

**Figure supplement 6—source data 1.** Source data for *Figure 2—figure supplement 6A–C*.

**Figure supplement 7.** Tracking dynamics of KcsA* potassium channels under phytohormone rhythms.

**Figure supplement 7—source data 1.** Source data for *Figure 2—figure supplement 7D–G*.

**Figure 2—video 1.** Dual-feedback gene circuit shows coordinated oscillations of fluorescence marker.

https://elifesciences.org/articles/78075/figures#fig2video1

**Figure 2—video 2.** Dynamics of KcsA-EGFP fluorescent reporter in the dual-feedback circuit show coherent oscillations under 3-hour cycles of phytohormone stimuli related to Figure 2—figure supplement 7.

https://elifesciences.org/articles/78075/figures#fig2video2

To test initially our computer model, we implemented a synthetic feedback circuit composed of IacR tagged with herpes simplex virus trans-activation domain (VP64) (*Hagmann et al., 1997*), MarR tagged with repression Mig1 silencing domain (*Ostling et al., 1996*; *Figure 2A*, *Figure 2—figure supplement 2*), and unstable dEGFP fluorescent reporter (*Dantuma et al., 2000*). All three components of the circuit were placed under control of the same synthetic promoter carrying MarR and IacR operator sites (*Pérez-García et al., 2021*). We found that this dual-feedback circuit responds sharply to SA–IAA gradient and shows excitable-like dynamics characterized by transient response peak in the absence of periodic hormone stimulus (*Figure 2—figure supplements 3 and 4*). Furthermore, we confirmed that the long step-like stimulation with IAA in dynamic microfluidic setup resulted in refractory dynamics of dEGFP signal after IAA removal (*Figure 2C*), characterized by transient response peak similar to that observed in the static environment (*Figure 2—figure supplement 4*) and in our computer model simulations (*Figure 2B*).

Next, we impose cyclic applications of IAA and SA with defined periods to guide gene expression patterns across yeast colonies. Spatially coordinated pulses of dEGFP were observed under various phytohormone stimulations of 2 hr, 1 hr, and 30 min (*Figure 2E, F*, and *Figure 2—figure supplement 5*, *Figure 2—video 1*) which are in a good agreement with computer model simulations (*Figure 2D*). The coordinated gene expression was confirmed by the analysis of cumulative dEGFP signal autocorrelation (*Figure 2G*), period distribution (*Figure 2H*, *Figure 2—figure supplement 6A*), complemented by peak characteristic analysis (*Figure 2—figure supplement 6B, C*). Importantly, we quantified that ~95% of all communities show coordinated peaks of dEGFP fluorescence (*Figure 2I*). Finally, to confirm that phytohormone changes directly control ion channel dynamics, we tested whether observed dynamics of circuit actually control of the ion channel expression. For that we fused KcsA* potassium channel (*Figure 1B*) with EGFP to monitor directly the life span of the channel under control of a feedback circuit. Indeed, phytohormone stimuli lead to coordinated spikes of KcsA-EGFP fluorescence mimicking that of dEGFP fluorescent marker, indicating that our circuit specifically controls the amount of heterologous ion channels inside the cell under guidance of phytohormones (*Figure 2—figure supplement 7*, *Figure 2—video 2*).

In summary, this chemically excited circuit presents a plausible regulatory module for engineering macroscopically coordinated ion channel expression in yeast cell populations.

## Coupling feedback circuit to PMP changes at the population level by engineered potassium channels

We have demonstrated that dual-feedback circuit shows characteristics of excitable system that robustly controls the ion channel expression in yeast cell collectives. Next, we tested how these changes in ion channel presence would reflect upon PMP changes by measuring ThT cationic dye fluorescence dynamics. In particular, we asked if the close-loop system (*Figure 3A*) would outperform our open-loop system (*Figure 1A*).

Next, we grew engineered yeasts carrying a dual-feedback circuit in the microfluidic device that was subjected to different frequencies of SA and IAA stimuli (3, 2, and 1 hr, respectively). We could observe cyclic bursts of ThT fluorescence that were remarkably consistent across cell populations in long-term experiments (*Figure 3A, B*, and *Figure 3—videos 1–3*). Cumulative autocorrelation and synchronicity analyses confirmed further the robustness of temporal response on the macroscopic scale (*Figure 3C, D*). The faster pace of environmental changes led to somehow less robust response possibly due to toxic effect of upregulation ion channels in short time periods (*Figure 3D*, *Figure 3—figure supplement 1*, *Figure 3—video 3*). Interestingly, while peak width and period were very consistent between different yeast colonies, we observed visible variation in amplitudes of ThT fluorescence (*Figure 3E*). These could be due to subtle differences in relative levels of KcsA* channels produced by each cell as observed in our experimental data (*Figure 2—figure supplement 7*) but also to inherent noise nature of gene expression. Nevertheless, our experiments indicate that dual-feedback integration in our original open-loop circuit significantly improves the robustness of ion channel control, resulting in coordinated modulation of ion channel expression and consequently organized changes of PMP across eukaryotic cell communities.

The PMP of yeast is regulated through voltage-gated potassium channels such as the outward-rectifier channel TOK1 (*Martinac et al., 2008*; *Mackie and Brodsky, 2018*). TOK1 is the main target for several toxins and volatile anesthetic agents (*Ahmed et al., 1999*), which cause uncontrolled opening and leakage of potassium ions to the extracellular space. Overexpression of TOK1 causes membrane hyperpolarization (more negative PMP) while *tok1* mutation leads to membrane depolarization often accompanied by cell death (*Sesti et al., 2001*). Thereby, TOK1 controls potassium release from yeast cells and maintains a balance in the PMP. Next, we checked if our dual-feedback circuit could be plugged into the regulation of this native TOK1 potassium channel in yeast. This strategy would potentially allow to plug a synthetic circuit to any native potassium channels in eukaryotes.

For that purpose, we constructed TOK1* channel by tagging degron domain at c-terminus of TOK1 to decrease half-life of the channel and thus increase its dynamics similar to that of KcsA* construction. Then we plugged our feedback circuit to control TOK1* expression levels on the macroscopic level (*Figure 3F*). We recorded ThT fluorescence changes over time under pulses of IAA and SA. Similarly, to KcsA* we observed consistent ThT pulses across number of independent yeast colonies (*Figure 3G, H*, *Figure 3—video 4*) as exemplified by persistence of pulsing period and synchronicity (*Figure 3—figure supplement 2*). Again, the autocorrelation analysis revealed a dominant pattern of ThT changes monitored across yeast colonies (*Figure 3—figure supplement 2*). These data indicate that our dual-feedback circuit can indeed control also the expression of native channels to modulate its activity by controlling the number of TOK1* transcripts available in the cell. Therefore, our findings highlight a generality of strategy that includes engineered dual-feedback system for controlling potassium channels and PMP through environmental rhythms.

## Concluding remarks

In the last decade, many efforts have been invested into developing new methods for the local control of ion channel activity with light or chemical signals. In contrast, mechanisms controlling ion channel expression have received substantially less attention, despite their importance in cardiac, neurological disorders and various forms of cancer. Here, we demonstrate a synthetic biology model for the control of potassium channel expression on macroscopic scales in eukaryotic cell populations. As a proof-of-concept, we tested this chemically driven mechanism in yeast to access the general suitability of our strategy for tuning cell electrophysiology at the population level. We demonstrated that a synthetic gene circuit controls ion channel dosage in each individual cell that is dictated by environmental input with selective frequencies. Changes in ion channel expression correlate with changes of PMP as shown by cationic dye translocation to the cell interior with increased fluorescence. Both heterologous

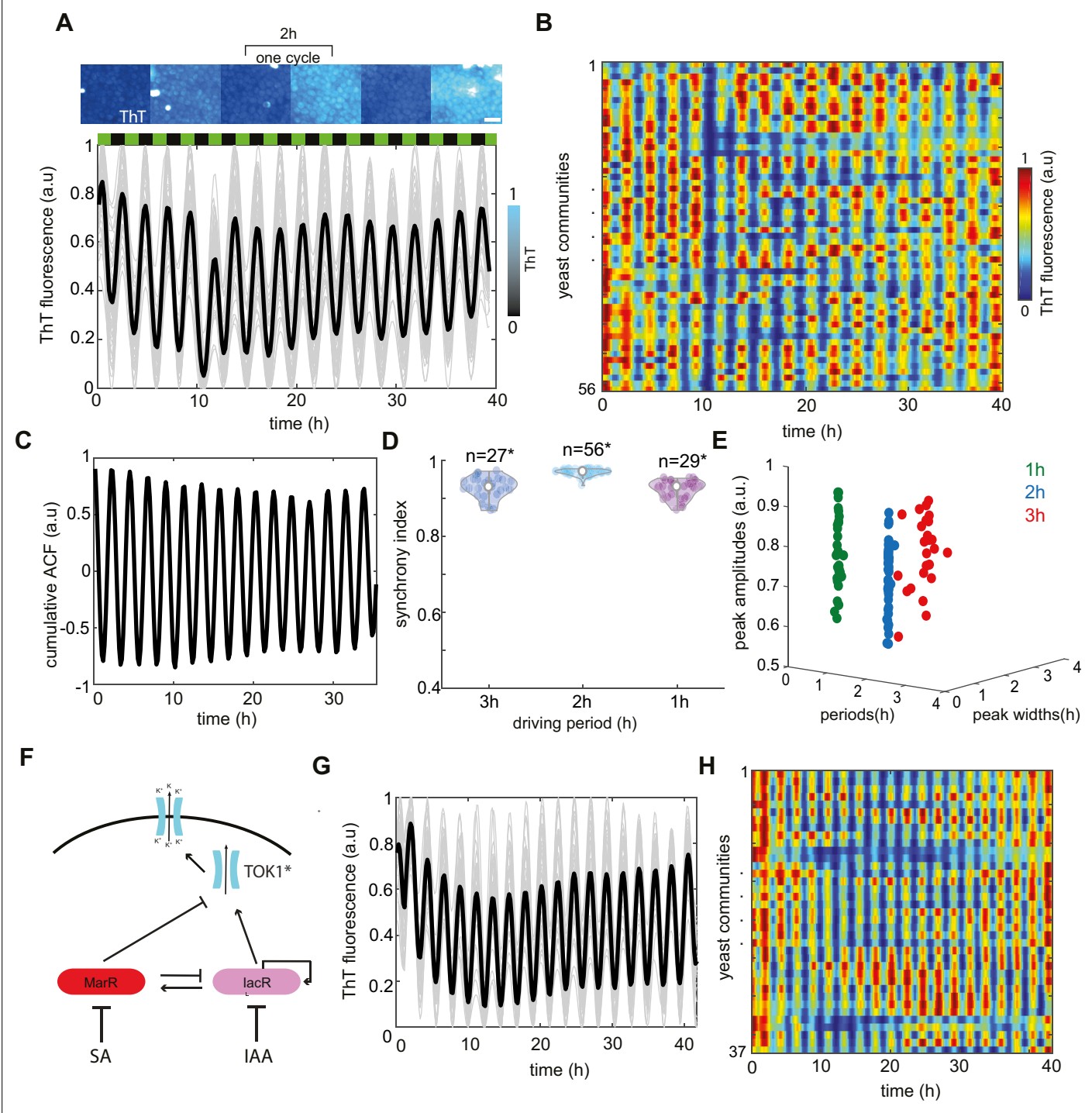

**Figure 3.** Global modulation of ion channel expression and plasma membrane potential (PMP) in yeast communities through phytohormones. (**A**) Representative time traces of Thioflavin T (ThT) fluorescence per community (n = 56 yeast communities) under 2-hr hormone stimuli. Average trend is shown in black. SA and IAA peaks are shown as black and green boxes above the traces. Scale bar represents 20µm (**B**) Kymograph of ThT fluorescence equivalent to time traces (**A**) with color coded map. (**C**) Cumulative autocorrelation analysis for all n = 56 communities show clear periodic, synchronized ThT fluorescence changes across communities. (**D**) Synchrony index calculated for each of the communities for three different frequencies of phytohormone stimuli (1, 2, and 3 hr) show good agreement with autocorrelation analysis. (**E**) Variation in amplitudes of ThT fluorescence plotted against periods of ThT and ThT peak widths. (**F**) Schematic of dual-feedback circuit controlling the yeast potassium channel TOK1*. Time-lapse ThT fluorescence changes in TOK1* integrating synthetic circuit (**G**) and corresponding heat map (**H**). Violin plots represent medians (white dots), interquartile zones (gray bars) and 95% confidence levels (solid gray line).

*Figure 3 continued on next page*

*Figure 3 continued*

The online version of this article includes the following video, source data, and figure supplement(s) for figure 3:

**Source data 1.** Source data for *Figure 3D, E*.

**Figure supplement 1.** Characterization of synchronicity of Thioflavin T (ThT) fluorescence in dynamically changing environment.

**Figure supplement 1—source data 1.** Source data for *Figure 3—figure supplement 1F–H*.

**Figure supplement 2.** Synchronicity analysis of TOK1*-based circuit controlling plasma membrane potential (PMP) in yeast communities.

**Figure supplement 2—source data 1.** Source data for *Figure 3—figure supplement 2B–E*.

**Figure 3—video 1.** Thioflavin T (ThT) oscillations controlled by phytohormone rhythms (2-hr stimuli) for yeast transformed with KcsA*-based circuit.
https://elifesciences.org/articles/78075/figures#fig3video1

**Figure 3—video 2.** Thioflavin T (ThT) oscillations controlled by phytohormone rhythms (3-hr stimuli) for yeast transformed with KcsA*-based circuit.
https://elifesciences.org/articles/78075/figures#fig3video2

**Figure 3—video 3.** Thioflavin T (ThT) oscillations controlled by phytohormone rhythms (1-hr stimuli) for yeast transformed with KcsA*-based circuit.
https://elifesciences.org/articles/78075/figures#fig3video3

**Figure 3—video 4.** Thioflavin T (ThT) oscillations controlled by phytohormone rhythms (2-hr stimuli) for yeast transformed with TOK1*-based circuit.
https://elifesciences.org/articles/78075/figures#fig3video4

and native channels were used as effectors of dual-feedback system to control PMP in yeast cells. Whereas the lack of feedback causes a noisy control of PMP, highlighting strong benefits of using dual-feedback regulation for the robust PMP control.

This work opens new avenues in the field of synthetic biology and cellular engineering by enabling a transformation of clocked chemical cues into coordinated ion channel expression without need for apparent cell-to-cell coupling. Our work proposes that coherent changes of ion channel expression can adjust electrical activity in growing cell populations and on extended spatial–temporal scales. Furthermore, our circuit could be potentially plugged into native context where timed processes would control circuit components endogenously and connect circuit output to native ion channel dosage. Nevertheless, our initial, proof-of-concept study produces some variability in response amplitudes. Therefore, future studies are needed to establish systems with improved response robustness to timed inputs, potentially leading to the creation of powerful toolboxes for controlling macroscopic electrophysiology across the tree of life. Practically, controlling locally and globally ion channel expression holds a key to designing more effective therapies providing the control of electrical status of abnormal cells in a broad spectrum of diseases.

## Materials and methods
### Strains and plasmid constructions

Constructs were cloned using isothermal Gibson assembly cloning. A middle-copy (~10–30 copies) episomal plasmid pGADT7 (Takara Bio Inc) was used to increase the concentration of proteins to buffer for the effects of intrinsic molecular noise and selected using different auxotrophic selection markers (Leucine, Uracil, and Histidine). $Iacr_o$/$MarR_o$ promoter and either standard CYC1 or ADH1 yeast terminators were cloned into activator or repressor plasmids (*Figure 1—figure supplement 1*). MarR, IacR, and KcsA* were codon optimized for yeast and synthesized using services delivered by Integrated DNA Technologies (IDT). The reporter plasmids include synthetic minimal promoters (synthesized with IDT) with previously identified MarR or IacR operator (*Alekshun et al., 2001*; *Shu et al., 2015*) sequences upstream TATA-box and minimal CYC1 promoter and fast-degradable UBG76V-EGFP (dEGFP) (*Dantuma et al., 2000*). KcsA bacterial potassium channel was engineered to include open configuration mutations as previously described (*Cuello et al., 2010*; *Sun et al., 2020*) and modified further to include plasma membrane localization signals and c-terminus mouse ODC degron signal (*Takeuchi et al., 2008*). KcsA* or TOK1* replaces the dEGFP gene on the reporter plasmid, respectively (*Figure 1—figure supplement 1* and *Figure 2—figure supplement 2*). PCR reactions were performed using Q5 high fidelity polymerase (New England Biolabs). Correct PCR products were digested with DpnI (New England Biolabs) to remove the template and subsequently cleaned up with a DNA cleanup kit (Zymo Research) before Gibson assembly. Constructs were transformed in ultra-competent cells from *E. coli* DH5a strain using standard protocols. All plasmids were

confirmed by colony PCR and validated with sequencing. The BY4741 laboratory yeast strain (a kind gift from Dr. Luis Rubio) carrying integrated copy constitutively expressed mCherry reporter was used to prepare competent cells and transformation of plasmids using Frozen-EZ Yeast Transformation II Kit (Zymo Research). DNA sequences used in this study are summarized in *Supplementary file 1*. Yeast strains are embedded in Key Resource Table.

## Multiwell plate and microscopy fluorescence measurements

Overnight culture of the yeast grown in 2% sucrose low fluorescence media (Formedium, UK) was diluted 100× and pipetted directly to a 96-well plate containing 2% sucrose and gradient of SA and IAA concentrations. Plates were incubated at 30° C overnight and well mixed by shaking before performing measurements. Measurements were done with the Thermo Scientific VarioskanTM LUX multimode microplate reader after 24 hr or were recorded every 10 min to generate a time-lapse profile of the dEGFP and $OD_{600}$. $OD_{600}$ was set at an absorbance of 600 nm wavelength, the fluorescence excitation and emission light at 488 and 517 nm wavelength for dEGFP. The PMP reporter activity was analyzed by measuring dye expression using potassium clamp experiments 24 hr after staining. Overnight cultures were diluted to a total $OD_{600}$ of 0.1. KCl at different concentrations was added to the diluted cultures. Aliquotes of 200 µl were pipetted from these diluted cultures to a multi-well plate containing different KCl concentrations (0, 50, 100, 200, and 400 mM) and 10 µM Thioflavin T, 10 µM DIBAC4(3), and 10 µM DIS-C3(3) for a direct comparison. Plates were incubated at 30°C overnight. The next day, each well was imaged in two different channels: differential interference contrast (DIC) and GFP ($\lambda_{Ex}$ = 488 nm; $\lambda_{Em}$ = 515 nm). The image acquisition was controlled by the software µManager and Leica DMI9 and images were captured using a ×10 dry objective (NA = 0.32).

## Time-lapse imaging, growth conditions, and data analysis

Live-cell imaging was performed on the Automated inverted Leica DMi8 fluorescence microscope equipped with Hamamatsu Orca Flash V3 camera that was controlled by Micro-Manager v.2.0 (https://micro-manager.org/). Images were captured with ×40 dry objective NA = 0.8 (Leica Inc). Traps containing cells were imaged every 10 min on three different channels (DIC, GFP Excitation: 488, Emission: 515, and mCherry Excitation: 583, Emission: 610) with CoolLed pE600 LED excitation source and standard Chroma epifluorescent filter set. Experiments were run for up to 72 hr under the continuous supply of nutrients in the microfluidic device. Acquired images were initially processed in Fiji 2.0 (https://imagej.net/Fiji) using custom scripting to extract positions with exponentially growing yeast cells. Constitutively expressed mCherry marker was used to identify exponentially growing cells and used to derive normalized dEGFP fluorescence: Dead or non-growing individuals were discarded by correcting dEGFP or KcsA-EGFP signal according to the formula *dEGFP/(dEGFP + mCherry)*. Each image was divided into 25 regions of interest and analyzed separately to isolate regions where cells were actively growing and could be tracked over time. The posterior analysis was done with custom R-studio scripts. Firstly, raw data were detrended using the detrend function from 'pracma' R-studio v4.0.3 package and then smoothed with Savitzky-Golay Smoothing function (savgol), from the same package, with a filter length of 15 was applied and the signal was normalized between 0 and 1 to generate heat maps across cell traps. Amplitudes were calculated with find peaks within the Process Data using the 'findpeaks' function from 'pracma' R package with nups and ndowns of 6, and periods were calculated by calculating distances between successive dEGFP peaks. Phase drift was calculated by comparing time differences of successive dEGFP peaks between cell communities in a microfluidic device to derive inter-community measures. Cumulative autocorrelations traces and power spectrum densities were calculated on mean dEGFP trajectories per colony calculated for *n* yeast communities (*n* > 20, ~10,000 cells each) using standard calculations with Matlab 2018b derived packages autocorrect and Fast Fourier Transformation (FFT). A quantitative metric of 'synchrony index' is defined as 1 − *R*, where *R* is the ratio of differences in subsequent ThT peak positions among cell communities (phase) to expected period. This metrics describes how well are yeast colonies synchronized with each other under guidance of the common environmental cue. The identical image analysis procedures were used to evaluate ThT fluorescence dynamics, autocorrelation function, and frequency responses.

## Mathematical model description

To derive a mathematical model of the excitable circuit (*Figure 2B* and *Figure 2—figure supplement 1*), we used a system of coupled ordinary differential equations adapted from previous theoretical studies (*Lindner et al., 2004*). Briefly, IacR and MarR protein concentrations change over time according to the following formulas:

$$\frac{\partial \text{IacR}}{\partial t} = a_1 + \frac{b_1 \cdot \text{IacR}^2}{K_A^2 + \text{IacR}^2 + (\gamma \cdot \text{MarR})^2} - d_1 \cdot (1 + \epsilon_I \cdot IAA) \cdot \text{IacR} \tag{1}$$

$$\frac{\partial \text{MarR}}{\partial t} = a_2 + \frac{b_2 \cdot \text{IacR}^2}{K_B^2 + \text{IacR}^2} - d_2 \cdot (1 + \epsilon_M \cdot SA) \cdot \text{MarR} \tag{2}$$

where $a_1$ and $a_2$ are basal IacR and MarR production rates. $b_1$ and $b_2$ are IacR-dependent protein production rates. $K_A$ and $K_B$ are half-max hill function coefficients, and $\gamma$ is the rate of MarR-dependent repression. $d_1$ and $d_2$ are protein turnover rates, respectively. $\epsilon_I$ and $\epsilon_M$ are the rates of phytohormone effect on a total protein turnover. IAA and SA are modeled using square wave and sine signal generators (Matlab Inc). The ratio of protein turnover $\frac{d_1}{d_2}$ (*Figure 2—figure supplement 1A–C*) represents a key bifurcation parameter that enables the transition of the system into oscillatory and excitable regimes. Model parameters are summarized in *Supplementary file 2*.

To calculate nullclines we set the left-hand side of *Equations 1 and 2* to 0 and setting $\epsilon_I$ and $\epsilon_M$ to 0. One can derive the following terms for calculation of phase portraits MarR*(IacR) and MarR**(IacR) (*Figure 2—figure supplement 1A–C*):

$$\text{MarR}^* = \frac{1}{\gamma} \cdot \sqrt{-\frac{K_A^2 \cdot a_1 + a_1 \cdot \text{IacR}^2 + b_1 \cdot \text{IacR}^2 - d_1 \cdot \text{IacR}^3 - K_A^2 \cdot d_1 \cdot \text{IacR}}{a_1 - d_1 \cdot \text{IacR}}} \tag{3}$$

$$\text{MarR}^{**} = \frac{a_2 + \frac{b_2 \cdot \text{IacR}^2}{K_B^2 + \text{IacR}^2}}{d_2} \tag{4}$$

## Cell loading procedure

All tubing lines were sterilized with ethanol and plugged into syringes or introduced in falcon tubes under sterile conditions. Fresh yeast colony was grown in low fluorescence media composition (Formedium, UK) with 2% sucrose as a carbon source or 2% glucose (K1 toxin-producing strain). The next day, yeast cultures were diluted 10–20 times approximately to 0.2–0.4 OD$_{600}$ to obtain highly concentrated cells that were transferred to a 50-ml falcon tube for loading. 60-ml media syringes were filled with 25 ml of inducing media (2% sucrose + 0.5% galactose) with or without compounds and 50-ml waste falcon tubes were filled with 10 ml of Distilled De-Ionized water. Before loading, devices were vacuumed for at least 20 min to remove all the air from the channels and traps. Syringes and falcon tubes were placed on the height control system and lines were connected as follows: media syringes were plugged first and kept above all other inputs to prevent media contamination. Adjusting the height of the cells containing falcon tube as well as media and waste aids in controlling the cell seeding in the traps. Although many cells pass through the chip directly toward the waste port, few cells got captured via microvalves and seeded the traps. Once 10–20 cells were captured in each trapping region, the flow from the cell loading port was reverted by decreasing the height to the same level as for the auxiliary waste.

## Microfluidic mold fabrication

Molds for the production of microfluidic device (*Pérez-García et al., 2021*) were designed in Inkscape and printed on plastic sheets with the monochrome laser printer at 1200 dpi resolution as described in *Pérez-García et al., 2021*. A density of Ink deposition was used to control the feature height. Plastic wafers were cut and transferred to the thermal oven set to 160°C to shrink by one-third of the original size, then baked again for 10 min to smoothen and harden the ink. Finally, molds were cleaned with soap, rinsed with isopropanol and DDI water and dried using a nitrogen gun, and secured with Scotch tape before use. Each cell trap is 500 × 500 × 7 µm size, hosting ~10,000 haploid yeast cells.

## Soft lithography

Molds were introduced in plastic 90-mm Petri dishes and fixed with double-sided tape. Dowsil Sylgard 184 Polydimethylsiloxane (PDMS) was properly mixed in a 10:1 (wt/wt) ratio of elastomer

and curing agent and stirred until the uniform consistency was achieved. Approximately 27 ml of the homogeneous mixture was poured into each Petri dish and completely degassed using the 8 CFM 2-stage vacuum pump for approximately 20 min. Degassed PDMS was cured at 80°C for 2 hr. Cured PDMS was removed from the Petri dish, separated from the wafer, and cut to extract the individual chips. Fluid access ports were punched with a 0.7-mm diameter World Precision Instruments (WPI) biopsy puncher and flushed with ethanol to remove any remaining PDMS. Individual chips were cleaned with ethanol and DDI water and Scotch tape to remove any remaining dirt particles.

## Microfluidic device bonding

At least 1 day before use, individual chips and coverslips were cleaned in the sonic bath and rinsed in ethanol, isopropanol, and water. Both surfaces were exposed to Corona SB plasma treater (ElectroTechnics Model BD-20AC Hand-Held Laboratory Corona Treater) between 45 s and 1 min, then surfaces were brought together and introduced at 80°C in an oven overnight to obtain the enhanced bond strength.

## Acknowledgements

We would like to thank Dr. Luis Rubio for providing BY4741 laboratory yeast strain used in this work. This work was supported by the Programa de Atraccion de Talento 2017 (Comunidad de Madrid, 2017-T1/BIO-5654 to KW), Severo Ochoa (SO) Programme for Centres of Excellence in R&D from the Agencia Estatal de Investigacion of Spain (grant SEV-2016-0672 (2017–2021) to KW via the CBGP). In the frame of SEV-2016-0672 funding MA received a PhD fellowship (SEV-2016-0672-18-3: PRE2018-084946). KW was supported by Programa Estatal de Generacion del Conocimiento y Fortalecimiento Científico y Tecnologico del Sistema de I+D+I 2019 (PGC2018-093387-A-I00) from MICIU (to KW). UPM Plan Propio Predoctoral fellow finances MGN.

## Additional information

### Funding

| Funder | Grant reference number | Author |
| --- | --- | --- |
| Comunidad de Madrid | Programa de Atraccion de Talento 2017-2023 2017-T1/BIO-5654 | Krzysztof Wabnik |
| Ministerio de Ciencia, Innovación y Universidades | PGC2018-093387-A-I00 | Krzysztof Wabnik |
| Agencia Estatal de Investigación | SEV-2016-0672 | Krzysztof Wabnik |
| Agencia Estatal de Investigación | SEV-2016-0672-18-3:PRE2018-084946 | Merisa Avdovic |
| Universidad Politécnica de Madrid | Plan Propio Predoctoral fellow | Mario García-Navarrete |

The funders had no role in study design, data collection, and interpretation, or the decision to submit the work for publication.

### Author contributions

Mario García-Navarrete, Data curation, Software, Formal analysis, Validation, Investigation, Visualization, Methodology, Writing - original draft; Merisa Avdovic, Data curation, Formal analysis, Validation, Investigation, Visualization, Methodology, Writing - original draft, Writing - review and editing; Sara Pérez-Garcia, Data curation, Software, Formal analysis, Validation, Investigation, Visualization, Methodology; Diego Ruiz Sanchis, Formal analysis, Investigation, Methodology, Writing - original draft; Krzysztof Wabnik, Conceptualization, Formal analysis, Supervision, Funding acquisition, Investigation, Writing - original draft, Project administration, Writing - review and editing

## Author ORCIDs

Mario García-Navarrete ⓘ http://orcid.org/0000-0002-1899-8206
Krzysztof Wabnik ⓘ http://orcid.org/0000-0001-7263-0560

## Decision letter and Author response

Decision letter https://doi.org/10.7554/eLife.78075.sa1
Author response https://doi.org/10.7554/eLife.78075.sa2

## Additional files

### Supplementary files

- Supplementary file 1. Oligos and synthetic DNA sequences used in this study.
- Supplementary file 2. Computer model parameters.
- MDAR checklist

### Data availability

All data are shown in the manuscript, figure supplements, or the supplementary files. Plasmids have been deposited to Addgene lab database https://www.addgene.org/browse/article/28233359/.

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

# Appendix 1

## Appendix 1—key resources table

| Reagent type (species) or resource | Designation | Source or reference | Identifiers | Additional information |
|---|---|---|---|---|
| Strain, BY4741 MATa his3Δ1 leu2Δ0 met15Δ0 ura3Δ0 (*Saccharomyces cerevisiae*) | sRedM | *Pérez-García et al., 2021* | BY4741 pGK1:: mCherry | Selection: KanMX (G418) |
| Strain, sRedM (*Saccharomyces cerevisiae*) | cLPdGFP | This study | sRedM pMarOIacO:: lacR-VP64 (pEX1004, ADDGENE_194950) pMarOIacO::MarR-RD (pEX1005, ADDGENE_ 194951) pMarOIacO::dEGFP (UbG76V-EGFP) (pEX1006, ADDGENE_194952) | Selection: KanMX (G418) -Leu, -Ura, -His |
| Strain, BY4741 MATa his3Δ1 leu2Δ0 met15Δ0 ura3Δ0 (*Saccharomyces cerevisiae*) | oLPKcsA* | This study | BY4741 MATa his3Δ1 leu2Δ0 met15Δ0 ura3Δ0 pGal7:: lacR-VP64 (pEX1001, ADDGENE_ 194713) pGal7::MarR-RD (pEX1002 ADDGENE_ 194714) pMarOIacO::KcsA*(pEX1003, ADDGENE_194949) | Selection: -Leu, -Ura, -His |
| Strain, BY4741 MATa his3Δ1 leu2Δ0 met15Δ0 ura3Δ0 (*Saccharomyces cerevisiae*) | sEmpty (control strain) | This study | BY4741 MATa his3Δ1 leu2Δ0 met15Δ0 ura3Δ0 Empty insert plasmids pGADT7 (Takara Bio) backbone with auxotrophic markers | Selection: -Leu, -Ura, -His |
| Strain, BY4741 MATa his3Δ1 leu2Δ0 met15Δ0 ura3Δ0 (*Saccharomyces cerevisiae*) | cLPKcsA* | This study | sRedM pMarOIacO:: lacR-VP64 (pEX1004, ADDGENE_194950) pMarOIacO::MarR-RD (pEX1005, ADDGENE_ 194951) pMarOIacO::KcsA*(pEX1003, ADDGENE_194949) | Selection: -Leu, -Ura, -His |
| Strain, BY4741 MATa his3Δ1 leu2Δ0 met15Δ0 ura3Δ0 (*Saccharomyces cerevisiae*) | cLPTOK1* | This study | pMarOIacO:: lacR-VP64 (pEX1004, ADDGENE_194950) pMarOIacO::MarR-RD (pEX1005, ADDGENE_ 194951) pMarOIacO::TOK1*(pEX1008, ADDGENE_194954) | Selection: -Leu, -Ura, -His |
| Strain, BY4741 MATa his3Δ1 leu2Δ0 met15Δ0 ura3Δ0 (*Saccharomyces cerevisiae*) | cLPKcsA-EGFP* | This study | sRedM pMarOIacO:: lacR-VP64 (pEX1004, ADDGENE_194950) pMarOIacO::MarR-RD (pEX1005, ADDGENE_ 194951) pMarOIacO::KcsA-GFP*(pEX1007, ADDGENE_194953) | Selection: KanMX (G418) -Leu, -Ura, -His |
| Other, PMP dye: Thioflavin T | ThT | Fisher Scientific, Thermo Scientific | CAS: 2390-54-7 | |
| Other, PMP dye: Bis-(1,3-Dibutylbarbituric Acid) Trimethine Oxonol | DIBAC4(3) | VWR INTERNATIONAL EUROLAB, S.L | CAS: 70363-83-6 | |
| Other, PMP dye: 3,3'-di-n-propylthiacarbocyanine iodide | DIS-C3(3) | FISHER SCIENTIFIC | CAS: 53336-12-2 | |

