## [Editor Report]

The important contribution of this study is the ability to leverage engineered gene circuits to control cellular membrane potential. The presentation of the data in this work is convincing and the controls are in place to demonstrate that electrophysiological changes arise from external chemical stimuli. This study will be of interest to those working on non-neuronal bioelectricity, particularly synthetic biologists and bioengineers.

---

## [Decision Letter]

**Decision letter after peer review:**

Thank you for submitting your article "Macroscopic control of synchronous electrical signaling with chemically-excited gene expression" for consideration by *eLife*. Your article has been reviewed by 3 peer reviewers, one of whom is a member of our Board of Reviewing Editors, and the evaluation has been overseen by a Reviewing Editor and Naama Barkai as the Senior Editor. The following individual involved in review of your submission has agreed to reveal their identity: Joseph Larkin (Reviewer #3).

Essential revisions:

1) The title/abstract should be refocused to de-emphasize communication/electrical signaling/synchronous, since the coordinated behavior is largely driven by external chemical stimuli. The mechanistic basis of the external chemical stimuli -> membrane potential change is the priority and is impactful enough on its own.

2) With that goal in mind, the rationale behind the excitable dynamics, as well as each of the molecular mechanistic steps, should be made clearer:

a) Why are excitable dynamics important here? This should be justified more clearly in terms of the synthetic biology application, as it makes implementation by other researchers more complex. Does the circuit as constructed truly exhibit excitable dynamics? This should be probed using their computational model along with further experiments. For example, by trying other non-periodic inputs or checking for a refractory period.

b) What is the clear causal chain of molecular events for producing a membrane potential change using chemical stimuli, including inducer uptake, excitable circuit activation, ion channel expression, ionic flux, membrane potential change, and finally membrane potential reporter fluorescence? This is the core contribution of the manuscript and is necessary for others to use this toolbox. This should be clearly illustrated with a schematic figure and each of these mechanistic steps should be tested with experimental controls. For example, by using a more standard voltage dye (such as TMRM or a voltage-sensitive protein) to track membrane potential, a K^+^ selective dye to track ionic flux, and matching degradation tags between KCsA and GFP to demonstrate the expression lifetime of the channel. Is it possible that membrane channels are more difficult to degrade than free proteins? Is it also possible that other ionic fluxes and/or cellular metabolism related to PMP could be interacting with the excitable circuit dynamics?

*Reviewer #1 (Recommendations for the authors):*

Overall, I think that this is a promising manuscript that should be accepted following minor experimental revisions to supplement the results with additional mechanistic controls (described below).

Specific Questions/Suggestions:

1) The authors should use a more standard voltage-sensitive dye (such as TMRM) or a voltage-sensitive protein. ThT can act as a voltage-sensitive dye due to its positive charge but is nonstandard

2) The authors should use a K+ dye to confirm that voltage changes are due to K+ flux

3) KcsA and GFP have different degradation tags. Can you show an equivalent expression in KcsA system to show it tracks the dynamics? What if KcsA is always present but some other cellular feedback (such as YMC) results in the dynamics? Is the yeast metabolic cycle contributing to these dynamics? How do the timescales compare and can YMC oscillations be observed in this setting?

4) The time histogram for periods is intuitive in Figure 1, why isn't that continued in Figures 2/3 instead using a power spectra? I think using the period histograms would give more confidence in the results.

5) Is the TOK1 channel constitutively expressed based on literature data or data from the authors? Is there a chelator of K1 toxin to confirm its role as a relevant diffusible signal? Or pulsing external K1 toxin and/or other channel blocker?

6) Scales bars and scale information would be useful, particularly for the colony experiments. Additionally, images in Figure 3b could be improved, and (along with movies) suggest heterogeneity of voltage response and toxicity of KcsA expression and K1 toxin in general. How long can this strategy be sustained due to toxicity? Is this a challenge to bioengineering/synthetic biology applications?

*Reviewer #2 (Recommendations for the authors):*

Overall, I think this is good work that should be published in *eLife*, as many in the community could benefit from novel approaches to synthetic biology – much more needs to be done in this emerging area. I do believe that the authors have supported their claim of control of synchronous electrical signaling via channel expression. However, overall, the presentation can be improved in a way that leads to a clear comprehension of what has been achieved. While synchronous fluorescence is achieved, what does this mean, that Vmem is identical across cells? Or just that Vmem is changing at the same rate? Or is it synchronous gene expression? It's not explicitly made clear but should be the key part of the introduction or methods. The greatest addition that can be made is a clear causal chain in Figure 1 drawing out the steps of channel expression, channel function, Vmem change, fluorescence, etc. If the goal is truly a tool or toolbox for others to use, this is necessary. Furthermore, the limited rigor in which the electrical activity was characterized, and the light discussion on drawbacks/limitations, reduces the impact of the claim that it is a 'robust synthetic transcriptional toolbox'. I think this is very nice work but needs to be presented a bit more thoughtfully.

General Remarks:

1. I would like to see an experimental null model that is not simply a control population (Sup 2-2, B), but where chemical stimulants are delivered in a manner seeking to abolish the periodicity.

2. In Figure 3D, it seems that this K1->TOK is slightly less reliable than the previous 2 experiments. There are a couple of communities that don't seem to sync as much. Why? This should be discussed.

3. It seems that the shorter the cycle, the less reliable the method (see figure S1, 5). I didn't see this mentioned anywhere.

4. In keeping with (2) and (3), there is little discussion of drawbacks/limitations/etc. – please add.

5. I'm not sure of the difficulty of the experiments, but many times you list that each experiment has been repeated, 'at least two times.' Why not give a precise sample size? N = 2 seems low, and perhaps the authors want to state what the limitations/difficulties were (which in turn bears on the issue of this being a toolbox – people need to know how many N's are reasonable).

6. While phase difference is a fine measure, there are many ways that periodic signals can be analyzed (wave shape, amplitude, etc.) including other measures of synchronicity. It may be useful to measure/characterize other aspects of how these electrical signals are related. I think this may be useful, as in Figure 2B and 3C that the mean dark line doesn't well represent the data spread.

Line Remarks:

1. Line 50-51: while I agree that ion expression may be noisy, it may also be attributed to biological degeneracy. It would be interesting to address this and how it may affect the results.

2. Line 69-71: I do not know why you chose Mar receptors, why it matters, the upsides/downsides, etc. Due to the earlier claim that this is a 'toolbox', please say more about these choices and what other choices could be made. As it stands, this is a single 'tool'.

3. Line 114-115: Does anything else contribute to potassium release? Are there any other mechanisms by which the PMP remains balanced? How does your method affect these, if at all?

4. Line 179-181: I do not understand the claim that this methodology is non-invasive. How would I do this in-vivo – don't you need a way to stimulate cells with chemicals in a periodic fashion?

5. Line 186-187: I would like to see in the discussion the author's thoughts on how this may disrupt electrical communication. In neuroscience, for example, electrical signaling is paramount for proper brain function. Would any system that depends on timed electrical communication not be eligible for this method?

Figure Remarks:

Overall, I think the figures need a bit more work and care put into them. They do not always communicate the ideas clearly, which is a shame given the valuable work:

1. Figure 1 – why is there a cyan channel inside the cell – what does this indicate?

2. Figure 1 – Most critical is to add a 'flow diagram' to walk me through what is happening overall. Figure A-B leave too much for my imagination. Especially for someone not familiar with the subject. Specifically, the causal chain downstream of the chemical stimulation – what happens next to the channel, the PMP, and eventually fluorescence – an explicit block diagram (and text) of what's driving what in this circuit.

3. Figure 1 (and others) the tiny boxes above C with SA/IAA are not obvious to see, nor to what they're doing. Again, more care should go into explaining the method and the results, as presenting this methodology is the entire point of the paper.

4. Figure 1D could have 'flow arrows' that better describe what's happening.

5. Figure 1 – The heat map is not labeled on the Y axis, and you reference specific community numbers a couple of times.

6. Figure 1E – The dashed colored lines make this too hard to read.

7. Figure S1, 3-4: These graphs are directly comparable, but have flipped y-axis. Why?

8. Figure 2E – I may be wrong, but the PSD seems strange. The peak of the dotted lines is ~0.002 Hz, which is around 8.3 minutes. However, this a 1h induced period. Is this off by a factor of 10? 0.0002 is closer to 1.3 hours.

9. Figure 2E – The entire point of this graph is to show that you can make a signal with a given frequency. However, I have no way to know what those peaks are, because they are not labeled, and the x-axis is making readers guess.

10. Figure S1, 6 A – please mark peaks or give me a x-axis that lets me guess better.

11. Figure S1, 6 D – I would not consider the variation here low. In fact, the title of the figure seems misleading. While yes, there is little change across stimulation period/shape the actual values are quite variable.

Small typos:

1. Line 37, 'which in turn provides (a?) power reservoir'

*Reviewer #3 (Recommendations for the authors):*

We would like to reiterate that this paper impressed us and we are enthusiastic about it.

First, here are some suggestions for addressing the two major issues we mentioned;

– To address this first issue, we think it is possible to remove references to signaling or communication within the text and focus it on chemical control of membrane potential. Again, we think that result in itself is impactful. The text and figures should make it clearer that the data show a group of cells all independently responding to the same driving stimulus. This is not engineering communication. It is a step toward that goal.

Another option would be for the authors to present analysis of the existing data that demonstrates spatial signal propagation.

We do not think Figure 2 supplement 3 should be included in the paper unless there is clear observation of a spatially propagating excitable signal.

– We suggest multiple approaches to argue for excitable dynamics. First, the authors could experimentally test several predictions of the excitable model with the microfluidic system. Do they observe a refractory period? Do they observe the expected behavior from the model if only one of the phytohormones is added or taken away? Supplements 3 and 4 of Figure 1 provide some support, but those results are not compared to specific predictions or a non-excitable scenario.

We have several overall questions and suggestions:

– Please describe the device in more detail. How physically large is each well? Roughly how many cells are contained in each well? When reporting average fluorescence values from colonies, roughly how many cells are being averaged over?

– The text often remarks about noise and how the system buffers noise. However, the Figure 1 video shows notable heterogeneity in GFP expression. Some cells have low signal, others very high. Is this expected for the excitable circuit? At the same time, the ThT movies from Figure 2 appear less heterogeneous, which is interesting given that the experiments have the same underlying circuit. Is this due to some buffering of noise by physiology that maintains membrane potential? Could it be due to buffering of cells by each other when they all release or take up potassium? What do the authors think about this? Or are we wrong about our observations of heterogeneity? The text presents no analysis, so one can only guess by looking at the movies.

– As described above, is it possible to perform a co-culture experiment of wild-type cells with the engineered KcsA* strain and drive the engineered strain with chemical stimuli? This would result in collective potassium leak by the engineered cells. Figure 2 supplement 1 suggests that this may modulate the membrane potential of the WT cells. While similar to the experiments of Figure 3, it may come closer to demonstrating electrical communication.

– The early discussion of TOK1 was distracting. We believe that TOK1 can be introduced with Figure 3.

– What do we know about the relevance of membrane potential in yeast? Given what we know, does this system offer any way to control yeast physiology? If the authors have any thoughts on this, it would be great to include those in the concluding remarks.

There are some components of the paper that were highlighted, but we didn't fully grasp their importance. It would be great if the authors could describe the importance of these aspects more. Here are the components whose importance we would like to better understand:

– Why is construction of an excitable circuit central to this result? Reasons to do this would be to synchronize cells and to create a spatially propagating wave. However, as we have indicated, it does not appear in the data that the system does these things.

– What is the importance of the phase drift measurements? Does the different phase drift for different stimulation patterns tell us something about the synthetic circuit?

We have several comments on the figures:

– Figure 1A and 1B are confusing. Figure 1A shows control of ion channels, which is the point of the paper, but not of Figure 1. This sets up the expectation that the results of Figure 1 are with ion channels. Figure 1B is very difficult to read. Perhaps color-coding the regulatory arrows for the two parts of the circuit would make it more clear? Or showing a simplified version like that of Figure 2A? As is, it takes a lot of examination and thought to understand what Figure 1B is showing.

– Is it possible to show where the pulses of the phytohormones are happening on the time trace graphs as shading in the background throughout the time trace? As the figures are now, it is difficult to tell that the chemical stimuli are periodic.

– In the autocorrelation graphs, why is one curve a heavy black line and the others light, colored, dotted lines? This makes it difficult to read the colored lines and leads the reader to believe there is something fundamentally different about those conditions from the black line.

– A small comment: is it possible to use a different color scale for ThT and GFP heatmaps? Or add color bar scales to the heatmaps with labels like "GFP Intensity" or "ThT Intensity"?

We believe some panels in the supplements could be brought into the main figures:

– Figure 1 – supplement 1B and D, could be added to main text Figure 1 to illustrate the excitable dynamics of the circuit.

– Figure 2 supplement 2A and B are essential and support what we believe is the most impressive result here, engineering the ability to dynamically control cellular membrane potential. Perhaps ACFs could be computed and compared for the two examples in this supplementary figure also.

---

## [Author Response]

Essential revisions:1) The title/abstract should be refocused to de-emphasize communication/electrical signaling/synchronous, since the coordinated behavior is largely driven by external chemical stimuli. The mechanistic basis of the external chemical stimuli -> membrane potential change is the priority and is impactful enough on its own.

We thank all three Reviewers for all constructive suggestions that with no doubt helped us to improve the original manuscript. We concur that coordinated behavior is guided by external stimuli in our systems, however, individual communities synchronize with each other by maintaining the low phase drift over time (guided by phytohormone stimuli). This synchronization mechanism has noise-filtering capacity, and it is based on Mar proteins (Perez-Garcia et al., Nat Comm, 2021). On contrary, classically driven systems build in bacteria (i.e., TetR, AraC, LacI) are limited by increasing phase drift which eventually leads to desynchrony among individual cells and populations (i.e., Mondragón-Palomino et al., 2011, Science). This is a key distinction between these classic driven systems and the coordinated system guided by external stimuli presented in our work.

In the revised version of the manuscript, we also demonstrate that the incorporation of feedbacks provides significantly more robust solution for coordinated expression of ion channels than that of the open-loop system.

Generally, we have substantially revised the manuscript including its structure, key message and title by focusing on the main finding; the coordinated regulation of ion channel expression that allows controllable modulation of PMP at the macroscopic scale in the eukaryotic model system.

2) With that goal in mind, the rationale behind the excitable dynamics, as well as each of the molecular mechanistic steps, should be made clearer:a) Why are excitable dynamics important here? This should be justified more clearly in terms of the synthetic biology application, as it makes implementation by other researchers more complex. Does the circuit as constructed truly exhibit excitable dynamics? This should be probed using their computational model along with further experiments. For example, by trying other non-periodic inputs or checking for a refractory period.

In the revised version we added several new experimental datasets. For instance, we tested the open-loop system (revised Figure 1) against the closed-loop system which incorporates positive and negative feedbacks (revised Figures 2 and 3). This additional control revealed the importance of circuit architecture for translating external stimuli into coordinated cellular behavior. We demonstrate that open-loop system can control ion channel expression but it seems considerably less robust then the closed-loop system when it comes to controlling changes in PMP which now has been discussed in the revised version (lines 107-120). Positive and negative feedbacks allow noise filtering and increases the responsiveness of the system which is reflected in coordinated PMP changes in yeast communities (revised Figure 3). Furthermore, we tested putative excitability using a long-term pulse of auxin. We found that our system could mimic the refractory dynamics seen in other excitable systems and our system was not able to respond to the subsequent pulse of phytohormone (Figure 2B and 2C, lines 138-149). Consistently, we observed a pulse of reporter response in the static environment (Figure 2—figure supplement 4). These observations suggest there may be a putative excitability that is an intrinsic property of our closed-loop system.

b) What is the clear causal chain of molecular events for producing a membrane potential change using chemical stimuli, including inducer uptake, excitable circuit activation, ion channel expression, ionic flux, membrane potential change, and finally membrane potential reporter fluorescence? This is the core contribution of the manuscript and is necessary for others to use this toolbox. This should be clearly illustrated with a schematic figure and each of these mechanistic steps should be tested with experimental controls. For example, by using a more standard voltage dye (such as TMRM or a voltage-sensitive protein) to track membrane potential, a K^+^ selective dye to track ionic flux, and matching degradation tags between KCsA and GFP to demonstrate the expression lifetime of the channel. Is it possible that membrane channels are more difficult to degrade than free proteins? Is it also possible that other ionic fluxes and/or cellular metabolism related to PMP could be interacting with the excitable circuit dynamics?

As suggested, we have now improved presentation of system components in Figures to clarify sequence of molecular events. We would also like to stress that the systems presented in this work is a first step towards control of ion channel expression and PMP in eukaryotes, as thus should not be considered as a ‘ready to use’ toolbox as certainly more work is required to establish such proper toolbox with exchangeable components and tunable dynamics. The following revisions and experiments were added to the corrected version of the manuscript:

1) Open-loop synthetic circuit integrating MarR and IacR controls expression of KcsA channel and this circuit is sensitive to both IAA and SA (revised Figure 1). Closed-loop synthetic circuit controlling heterologous KcsA* or native TOK1* channels, respectively (Figures 2 and 3).

2) Cyclic changes in ion channel expression increases or decrease potassium channels expression (Figure 1A) with is typically associated with change in PMP, for instance the overproduction of potassium channels cause membrane hyperpolarization while channels mutants cause PMP depolarization (i.e., Ahmed, A. et al. 1999, Cell; Mackie, T. D. and Brodsky, J. L, 2018, Genetics; Sesti, F et al., 2001, Cell).

3) We found that commonly used plasma membrane potential dyes used in yeast (i.e. Peña et al., 2020), such as DIBAC4(3) and DIS-C(3)3 are inferior to ThT in terms of signal strength, stability, hydrophobicity and linearity (dynamic range) of potassium change detection (Figure 1—figure supplement 2, lines 97-103). We did not use TMRM as this was used primarily to monitor mitochondrial membrane potential, and not plasma membrane potential which is outside of the scope of this work.

4) We have constructed KcsA-EGFP fusion construct maintaining ODC degradation tag; the same tag that used for both MarR and IacR. We demonstrate that KcsA* channel follows similar dynamics of that of dGFP or ThT fluorescence characterized by peaks of expression coordinated in a macroscopic context (Figure 2—figure supplement 7, main text lines 159165). Therefore, degradation tags that we used provide robust changes in the half-life time of both dGFP reporter and KcsA* ion channel. It is important to note that degradation tag used for the construction of dGFP is based on N-terminal degron rule and cannot be used for construction of KcsA as KcsA requires N-terminal signal peptide for the proper membrane targeting.

5) We constructed an engineered version of native yeast potassium channel (TOK1*) and we tested whether our circuit could control native voltage-sensitive channel by directly modulating TOK1 expression levels (Figure 3F, lines 209-220). We found that regulated expression of TOK1 provided organized changes in ThT fluorescence similar to those observed for synthetic KcsA*, indicating that control of ion channel dosage provides a more general strategy to control PMP dynamics.

6) We cannot completely exclude that cell metabolism or other ions could influence PMP changes in yeast. However, we have now tested three different periods of phytohormone stimuli (1h, 2h,3h) and we see coherent changes of ThT fluorescence that adapt to changes in ion channel expression. Furthermore, the response robustness was dependent on circuit architecture (open-loop versus closed-loop). In our view it is very unlikely some other metabolic processes or ions could provide such selective control of PMP changes that we observed in our engineered yeast populations.

7) We attempted to measure K^+^ flux using commercial ION Potassium Green-2 AM, K^+^ indicator (Abcam) in yeast which, to our knowledge, was never used before in yeast. However, we were unsuccessful in detecting any fluorescence even after control treatments with KCL. It is possible that dye is quenched or yeast does not have esterases to hydrolyse the AM. Again, to our knowledge this is the only fluorescent dye able to monitor K^+^ changes, thus more work is needed to establish robust K^+^ dyes in yeast before they could be used as robust track K^+^ changes in yeast. We feel this is currently outside of scope of this work.

Reviewer #1 (Recommendations for the authors):Overall, I think that this is a promising manuscript that should be accepted following minor experimental revisions to supplement the results with additional mechanistic controls (described below).Specific Questions/Suggestions:1) The authors should use a more standard voltage-sensitive dye (such as TMRM) or a voltage-sensitive protein. ThT can act as a voltage-sensitive dye due to its positive charge but is nonstandard

We tested several dyes including most commonly used PMP marker in yeast such as DIBAC4(3) and DIS-C3(3) (i.e. Peña et al., 2020) and we found ThT stands out in terms of stability, fluorescence strength, dynamic range, low hydrophobicity and absorption properties as compared to other dyes (Figure 1- figure supplement 2, and lines 97-103)., which is consistent with previous studies (i.e. Peña et al., 2020). TMRM was so far used to monitor static potential of mitochondrial membrane and thus is not suitable for long-term live imaging of plasma membrane potential. In early stage of this study, we indeed tested some voltage sensitive proteins i.e., ArcLight (Walrati Limapichat, et al., 2020) but signal intensities were marginal and dynamic range was very poor making these markers unsuitable for time-lapse cell imaging in microfluidic devices.

2) The authors should use a K+ dye to confirm that voltage changes are due to K+ flux

We attempted to measure K+ flux using commercial ION Potassium Green-2 AM, K+ indicator (Abcam) for first time in yeast. However, we were unsuccessful in detecting any fluorescence even after control treatments with KCL. It is possible that dye is quenched or yeast does not have esterases to hydrolyse the AM. To our knowledge this is the only dye able to monitor K+ changes, thus more studies is needed to establish working K+ dyes in yeast before they could be used as robust track K+ changes in yeast. We feel it is currently outside of scope of this work.

3) KcsA and GFP have different degradation tags. Can you show an equivalent expression in KcsA system to show it tracks the dynamics? What if KcsA is always present but some other cellular feedback (such as YMC) results in the dynamics? Is the yeast metabolic cycle contributing to these dynamics? How do the timescales compare and can YMC oscillations be observed in this setting?

We have constructed KcsA*-EGFP fusion construct maintaining ODC degradation tag that was also used for both MarR and IacR circuit components. We demonstrate that KcsA* channel follows similar dynamics of that of dGFP or ThT fluorescence characterized by peaks of expression coordinated in a macroscopic context (Figure 2- figure supplement 7, lines 159-166). Therefore, degradation tags that we used provide robust changes in the halflife time of both dGFP and KcsA* ion channel. It is important to note that degradation tag used for the construction of dGFP is based on N-terminal degron rule and cannot be used for the construction of KcsA as KcsA requires N-terminal signal peptide for proper membrane targeting.

4) The time histogram for periods is intuitive in Figure 1, why isn't that continued in Figures 2/3 instead using a power spectra? I think using the period histograms would give more confidence in the results.

In the revised manuscript, we have used violin plots to demonstrate distribution of periods in all experiments to ease comparison between different scenarios.

5) Is the TOK1 channel constitutively expressed based on literature data or data from the authors? Is there a chelator of K1 toxin to confirm its role as a relevant diffusible signal? Or pulsing external K1 toxin and/or other channel blocker?

We have included new experiments and remove K1 data from the manuscript as it may indeed confuse readers. Currently, we are unable to control K1 toxin half-life due to its extracellular, and middle-man nature. Instead, we decided to control TOK1 expression directly with our synthetic circuit (revised Figure 3). We constructed an engineered version of native yeast potassium channel (TOK1*) and we tested whether our circuit could control native voltage-sensitive channel by modulating TOK1 expression levels (Figure 3F-H, lines 198-220). We found that regulated expression of TOK1 provided organized changes in ThT fluorescence similar to those observed for synthetic KcsA*, indicating that control of ion channel dosage provides a more general strategy to control PMP dynamics.

6) Scales bars and scale information would be useful, particularly for the colony experiments. Additionally, images in Figure 3b could be improved, and (along with movies) suggest heterogeneity of voltage response and toxicity of KcsA expression and K1 toxin in general. How long can this strategy be sustained due to toxicity? Is this a challenge to bioengineering/synthetic biology applications?

We have included scale bars to the figures. Answering the toxicity question, yes, the toxicity is a recognizable challenge in synthetic biology applications, therefore it is critical to increase degradation rates of effectors to maintain low level of toxicity in such synthetic systems. This issue was one of main reasons for integrating degron tags into KcsA* and TOK1* in our study.

Reviewer #2 (Recommendations for the authors):Overall, I think this is good work that should be published in eLife, as many in the community could benefit from novel approaches to synthetic biology – much more needs to be done in this emerging area. I do believe that the authors have supported their claim of control of synchronous electrical signaling via channel expression. However, overall, the presentation can be improved in a way that leads to a clear comprehension of what has been achieved. While synchronous fluorescence is achieved, what does this mean, that Vmem is identical across cells? Or just that Vmem is changing at the same rate? Or is it synchronous gene expression? It's not explicitly made clear but should be the key part of the introduction or methods. The greatest addition that can be made is a clear causal chain in Figure 1 drawing out the steps of channel expression, channel function, Vmem change, fluorescence, etc. If the goal is truly a tool or toolbox for others to use, this is necessary. Furthermore, the limited rigor in which the electrical activity was characterized, and the light discussion on drawbacks/limitations, reduces the impact of the claim that it is a 'robust synthetic transcriptional toolbox'. I think this is very nice work but needs to be presented a bit more thoughtfully.

We do not measure Vmem directly, instead we use cationic dye that translocate in the event of PMP change. We see coordinated changes in ThT fluorescence which are of course the proxy for Vmem changes. But the underlying mechanism is the synchronized expression of ion channels. We show that changing dosage of ion channels in the cells may be sufficient to induce macroscopically organized changes in PMP. In the revised manuscript we improved and clarify sequence of events that led us conclude that coordinated changes in ion channels expression can control PMP on the extended spatial-temporal scale.

General Remarks:1. I would like to see an experimental null model that is not simply a control population (Sup 2-2, B), but where chemical stimulants are delivered in a manner seeking to abolish the periodicity.

Since our synthetic circuit responds directly to phytohormone changes it is expected that any unregular changes in stimuli will disrupt regular expression of ion channels. In our current flow setup, we are unable to generate chaotic unregular stimuli. However, we show that extending time of pulses can reveal underlying the refractory-like dynamics (Figure 2B, C). Importantly, we have included one more important control for testing influence of circuit architecture on the ion channel expression. In particular, we removed the feedback layers, creating open-loop systems and we demonstrate that it is substantially less reliable than its feedback-integrating circuit counterpart (please compare revised Figure 1 vs Figure 3).

2. In Figure 3D, it seems that this K1->TOK is slightly less reliable than the previous 2 experiments. There are a couple of communities that don't seem to sync as much. Why? This should be discussed.

We have included new experiments and remove K1 data from the manuscript as it may indeed confuse readers. In fact, we cannot control K1 toxin half-life due to its extracellular, middle-man nature, and toxicity effects are pronounced compared to a direct use of potassium channel such as KcsA*. Instead, we provided new data on the control of TOK1 expression with our synthetic circuit (see revised Figure 3) using an engineered version of native yeast potassium channel (TOK1*). We tested whether our circuit could control native voltagesensitive channel by modulating TOK1 expression levels. We found that regulated expression of TOK1 provided organized changes in ThT fluorescence similar to those observed for synthetic KcsA*, indicating that control of ion channel dosage may provide a more general strategy to control PMP dynamics (lines 209-220).

3. It seems that the shorter the cycle, the less reliable the method (see figure S1, 5). I didn't see this mentioned anywhere.

Shortest cycles of 30min are close to half-life time of fluorescence marker dGFP which is most likely the limiting factor for robust measurements of super rapid fluorescence changes. In fact, in our previous work, we show that the faster the cycle the more reliable system becomes (Perez-Garcia et al., 2021, Nat Comm) up to the limits of degradation machinery.

4. In keeping with (2) and (3), there is little discussion of drawbacks/limitations/etc. – please add.

We discuss limitations of this approach in the revised concluding remarks section.

5. I'm not sure of the difficulty of the experiments, but many times you list that each experiment has been repeated, 'at least two times.' Why not give a precise sample size? N = 2 seems low, and perhaps the authors want to state what the limitations/difficulties were (which in turn bears on the issue of this being a toolbox – people need to know how many N's are reasonable).

We have clarified the descriptions. n in many cases such as synchrony index is the number of analyzed communities (biological replicates). Each community consists of around 10000 cells (500um x 500um traps). Then, each experiment was repeated at least two time (technical replicates) on independent days. For instance, if we analyzed n=25* communities that means we took into account 250000 cells biological replicates x 2 technical replicates. For measurements of amplitudes, periods, peak widths we took all data for each of communities to present cumulative distributions. Therefore, we believe that represents statistically sounds numbers. We clarified description of biological and technical replicates in the revised manuscript.

6. While phase difference is a fine measure, there are many ways that periodic signals can be analyzed (wave shape, amplitude, etc.) including other measures of synchronicity. It may be useful to measure/characterize other aspects of how these electrical signals are related. I think this may be useful, as in Figure 2B and 3C that the mean dark line doesn't well represent the data spread.

Now we analyzed synchronicity using three different methods:

1) we calculate cumulative autocorrelations of response between communities.

2) To complement autocorrelation analysis, we developed a quantitative metric of

‘synchrony index’ defined as 1 – R where R is the ratio of differences in subsequent ThT peak positions amongst cell communities (phase) to expected period. This metrics describes how well synchronized are fungi colonies with each other under guidance of the common environmental signal.

3) we analyzed amplitudes and peak widths for all presented scenarios and we conclude that while periods and peak widths are robust across communities there is a noticeable variation in amplitudes (i.e., Figure 3E).

Based on those, we present multi-step rigorous approach to explore different aspects of oscillatory signal characteristics.

Line Remarks:1. Line 50-51: while I agree that ion expression may be noisy, it may also be attributed to biological degeneracy. It would be interesting to address this and how it may affect the results.

We thank Reviewer for this interesting comment.

To demonstrate importance of noise filtering, we show how the open-loop system that lacks noise filtering capacity (no feedback) performs much worse than its closed-loop counterpart in terms of PMP modulation on the macroscopic scale (compare Figure 1 and Figure 3). This indicates that a feedback-controlled ion expression is necessary to steer PMP in a coherent manner (lines 193-196).

2. Line 69-71: I do not know why you chose Mar receptors, why it matters, the upsides/downsides, etc. Due to the earlier claim that this is a 'toolbox', please say more about these choices and what other choices could be made. As it stands, this is a single 'tool'.

In revised manuscript we elaborated on the rationale for using Mar-based system that directly relates to our recent work (Perez-Garcia et al., 2021, Nat Comm) (lines 75-85). Again, we would like to stress that we propose a foundation for development of new tools or toolboxes for rational engineering of electrophysiology in eukaryotes. Therefore, this work should be treated as initial proof of concept rather than ‘ready to use’ modular toolbox, we concur more work is needed in the future to establish standard for this new approach (lines 244-250). We apologize for misunderstanding and adjust the text accordingly.

3. Line 114-115: Does anything else contribute to potassium release? Are there any other mechanisms by which the PMP remains balanced? How does your method affect these, if at all?

We are not aware of any other mechanism that could provide such selective regulation of PMP in yeast that we can control thorough environment. We believe that changes of PMP that we recorded are attributed to the architecture of synthetic circuit as we reporter significant differences in open-loop and closed-loop circuit variants. In particular feedback integrating circuits representation can robustly control ion channel expression thorugh environmental stimuli.

4. Line 179-181: I do not understand the claim that this methodology is non-invasive. How would I do this in-vivo – don't you need a way to stimulate cells with chemicals in a periodic fashion?

We apologize for the confusion. Now we removed that claim from the revised manuscript.

5. Line 186-187: I would like to see in the discussion the author's thoughts on how this may disrupt electrical communication. In neuroscience, for example, electrical signaling is paramount for proper brain function. Would any system that depends on timed electrical communication not be eligible for this method?

Our system could be integrated to control native ion channels to either disrupt or coordinate electrophysiology of cells. Furthermore, native system could be plugged to control timing of Mar protein dynamics to synchronize output with timed input as suggested by Reviewer. We included that speculation in revised manuscript (lines 238-250).

Figure Remarks:Overall, I think the figures need a bit more work and care put into them. They do not always communicate the ideas clearly, which is a shame given the valuable work:1. Figure 1 – why is there a cyan channel inside the cell – what does this indicate?

This has been clarified in the revised version of the manuscript.

2. Figure 1 – Most critical is to add a 'flow diagram' to walk me through what is happening overall. Figure A-B leave too much for my imagination. Especially for someone not familiar with the subject. Specifically, the causal chain downstream of the chemical stimulation – what happens next to the channel, the PMP, and eventually fluorescence – an explicit block diagram (and text) of what's driving what in this circuit.

This has been improved in revised version (Figures 1A and 2A and 3F).

3. Figure 1 (and others) the tiny boxes above C with SA/IAA are not obvious to see, nor to what they're doing. Again, more care should go into explaining the method and the results, as presenting this methodology is the entire point of the paper.

Amended in the revised version. We would like to comment that ‘toolbox’ creation is not the key point of this paper but rather the concept of using ion channel expression to control electrophysiology in cellular collectives.

4. Figure 1D could have 'flow arrows' that better describe what's happening.

Amended in the revised version of Figure 1C panel and legend.

5. Figure 1 – The heat map is not labeled on the Y axis, and you reference specific community numbers a couple of times.

Amended in the revised version.

6. Figure 1E – The dashed colored lines make this too hard to read.

Amended in the revised version.

7. Figure S1, 3-4: These graphs are directly comparable, but have flipped y-axis. Why?

These diagrams are not directly comparable. One is a heat map of SA-IAA concentration gradient (3) at given time point (entrance in stationary phase) while (4) has additional time component.

8. Figure 2E – I may be wrong, but the PSD seems strange. The peak of the dotted lines is ~0.002 Hz, which is around 8.3 minutes. However, this a 1h induced period. Is this off by a factor of 10? 0.0002 is closer to 1.3 hours.

Yes, we thank Reviewer for spotting this error it should be 10x less indeed, however we decided to not to use PSD diagrams in revised version to increase clarity and simplicity of analysis.

9. Figure 2E – The entire point of this graph is to show that you can make a signal with a given frequency. However, I have no way to know what those peaks are, because they are not labeled, and the x-axis is making readers guess.

It has been corrected in revised version.

10. Figure S1, 6 A – please mark peaks or give me a x-axis that lets me guess better.

This panel has been removed in revised version as all other PSD diagrams.

11. Figure S1, 6 D – I would not consider the variation here low. In fact, the title of the figure seems misleading. While yes, there is little change across stimulation period/shape the actual values are quite variable.

In revised version we removed CV analysis and focused on direct analysis of peak period, widths and amplitudes to clarify and simplify overall analysis of wave patterns and show cumulative distributions among all communities (space) and in time.

Small typos:1. Line 37, 'which in turn provides (a?) power reservoir'

Corrected accordingly.

Reviewer #3 (Recommendations for the authors):We would like to reiterate that this paper impressed us and we are enthusiastic about it.First, here are some suggestions for addressing the two major issues we mentioned;– To address this first issue, we think it is possible to remove references to signaling or communication within the text and focus it on chemical control of membrane potential. Again, we think that result in itself is impactful. The text and figures should make it clearer that the data show a group of cells all independently responding to the same driving stimulus. This is not engineering communication. It is a step toward that goal.

As suggested, we focused the story on coordinated regulation of ion channel expression and its impact on electrophysiology of individual cells in the collective. We removed connotation to communication or signaling between cells.

Another option would be for the authors to present analysis of the existing data that demonstrates spatial signal propagation.We do not think Figure 2 supplement 3 should be included in the paper unless there is clear observation of a spatially propagating excitable signal.– We suggest multiple approaches to argue for excitable dynamics. First, the authors could experimentally test several predictions of the excitable model with the microfluidic system. Do they observe a refractory period? Do they observe the expected behavior from the model if only one of the phytohormones is added or taken away? Supplements 3 and 4 of Figure 1 provide some support, but those results are not compared to specific predictions or a non-excitable scenario.

To explore putative excitability in our system we present two observations. Firstly, a transient peak of dGFP expression that was recorded in the static environment (Figure 2 —figure supplement 4) and has not been seen in our previous study that involve open-loop system (Perez-Garcia et al., 2021, Nat Comm). Secondly, we added new data when we stimulated cells with long 12h peak of auxin followed by second peak and revealed refractory-like dynamics characterized by lack of response capability to subsequent stimuli (Figure 2B and 2C). We believe these datasets provide further evidence towards existence of excitability in our dual feedback circuit.

We have several overall questions and suggestions:– Please describe the device in more detail. How physically large is each well? Roughly how many cells are contained in each well? When reporting average fluorescence values from colonies, roughly how many cells are being averaged over?

This description has been improved in revised version each trap/community contains ~10000 cells. Traps are approximately 500umx500umx7um.

– The text often remarks about noise and how the system buffers noise. However, the Figure 1 video shows notable heterogeneity in GFP expression. Some cells have low signal, others very high. Is this expected for the excitable circuit? At the same time, the ThT movies from Figure 2 appear less heterogeneous, which is interesting given that the experiments have the same underlying circuit. Is this due to some buffering of noise by physiology that maintains membrane potential? Could it be due to buffering of cells by each other when they all release or take up potassium? What do the authors think about this? Or are we wrong about our observations of heterogeneity? The text presents no analysis, so one can only guess by looking at the movies.

We believe dual feedback provide noise-filtering capabilities and responsiveness as compared to non-feedback alternative (see now revised Figure 1 and Figure 3). This is an eminent feature of putatively excitable system. We also show while periods and peak widths are very robust across communities, the amplitude may be quite variable (Figure 3E). Therefore, our circuit filters undesired frequencies but pass the noise in the amplitude of response (lines 193196, concluding remarks and various places in main text).

– As described above, is it possible to perform a co-culture experiment of wild-type cells with the engineered KcsA* strain and drive the engineered strain with chemical stimuli? This would result in collective potassium leak by the engineered cells. Figure 2 supplement 1 suggests that this may modulate the membrane potential of the WT cells. While similar to the experiments of Figure 3, it may come closer to demonstrating electrical communication.

As mentioned above, we have decided to revise structure of the story and remove connotation to electrical communication at present and focus on regulation of ion channel expression and non-coupled coordination of cell electrophysiology for a general clarity. We would prefer to explore this exciting possibility in the follow-up study.

– The early discussion of TOK1 was distracting. We believe that TOK1 can be introduced with Figure 3.

We rearranged and put more context to TOK1 in revised version (lines 198-220). By engineering TOK1 into our circuit (Figure 3F-H) we show that our concept could apply more generally to native potassium channels.

– What do we know about the relevance of membrane potential in yeast? Given what we know, does this system offer any way to control yeast physiology? If the authors have any thoughts on this, it would be great to include those in the concluding remarks.

In revised manuscript we discussed how this concept could be use more generally to control cell electrophysiology (lines 229-250).

There are some components of the paper that were highlighted, but we didn't fully grasp their importance. It would be great if the authors could describe the importance of these aspects more. Here are the components whose importance we would like to better understand:– Why is construction of an excitable circuit central to this result? Reasons to do this would be to synchronize cells and to create a spatially propagating wave. However, as we have indicated, it does not appear in the data that the system does these things.

In revised manuscript we discuss that putative excitability via dual feedback design could offer noise buffering capability supported by direct comparison with the open-loop system (Figures 1 and 3).

– What is the importance of the phase drift measurements? Does the different phase drift for different stimulation patterns tell us something about the synthetic circuit?

To remove confusions, we have analyzed synchronicity using following measures in the revised version:

1) we calculate cumulative autocorrelations of response between each community.

2) To complement autocorrelation analysis, we developed a quantitative metric of

‘synchrony index’ defined as 1 – R where R is the ratio of differences in subsequent ThT peak positions amongst cell communities (phase) to expected period. This metrics describes how well synchronized are fungi colonies with each other under guidance of the common environmental signal.

3) we analyzed amplitudes and peak widths for all presented scenarios.

All these metrics show that circuit is robust in terms of period and width of response but show variability in amplitudes. We do not see big changes in overall synchronicity for different stimuli frequencies (i.e. Figure 3D). However, we do see significant decrease of SI after removing feedback layer (Figure 1I) which tells us that circuit architecture is central to the performance of our system.

We have several comments on the figures:– Figure 1A and 1B are confusing. Figure 1A shows control of ion channels, which is the point of the paper, but not of Figure 1. This sets up the expectation that the results of Figure 1 are with ion channels. Figure 1B is very difficult to read. Perhaps color-coding the regulatory arrows for the two parts of the circuit would make it more clear? Or showing a simplified version like that of Figure 2A? As is, it takes a lot of examination and thought to understand what Figure 1B is showing.

We have simplified and clarified circuit diagrams in the revised version of figures to further guide readers.

– Is it possible to show where the pulses of the phytohormones are happening on the time trace graphs as shading in the background throughout the time trace? As the figures are now, it is difficult to tell that the chemical stimuli are periodic.

We provided pulses of phytohormones on the top of each time-course datasets (black box for SA and green box for IAA) in all adapted figures

– In the autocorrelation graphs, why is one curve a heavy black line and the others light, colored, dotted lines? This makes it difficult to read the colored lines and leads the reader to believe there is something fundamentally different about those conditions from the black line.

Color corresponds to different observable periods. Figure has been improved.

– A small comment: is it possible to use a different color scale for ThT and GFP heatmaps? Or add color bar scales to the heatmaps with labels like "GFP Intensity" or "ThT Intensity"?

Now, we provide separate heatmap labels for both dGFP and ThT.

We believe some panels in the supplements could be brought into the main figures:– Figure 1 – supplement 1B and D, could be added to main text Figure 1 to illustrate the excitable dynamics of the circuit.

This has been amended in revised Figure 1.

– Figure 2 supplement 2A and B are essential and support what we believe is the most impressive result here, engineering the ability to dynamically control cellular membrane potential. Perhaps ACFs could be computed and compared for the two examples in this supplementary figure also.

Now, we added critical datasets with cyclic stimuli for two different ion channels and two circuit architectures (open-loop (Figure 1) and closed-loop (Figures 2 and 3). Data on control condition have been also presented in revised Figure 1).